# Levels of and determinants for physical activity and physical inactivity in a group of healthy elderly people in Germany: Baseline results of the MOVING-study

**Fabian Kleinke**[1,2]*, **Peter Penndorf**[1,2], **Sabina Ulbricht**[2,3], **Marcus Dörr**[2,4], **Wolfgang Hoffmann**[1,2], **Neeltje van den Berg**[1,2]

**1** Institute for Community Medicine, Section Epidemiology of Health Care and Community Health, University Medicine Greifswald, Greifswald, Germany, **2** Partner Site Greifswald, DZHK (German Centre for Cardiovascular Research), Berlin, Germany, **3** Institute of Social Medicine and Prevention, University Medicine Greifswald, Greifswald, Germany, **4** Department of Internal Medicine B, University Medicine Greifswald, Greifswald, Germany

\* fabian.kleinke@uni-greifswald.de

**Data Availability Statement:** All relevant data are within the manuscript and its Supporting Information files.

## Abstract

### Background

Low levels of physical activity (PA) and high levels of physical inactivity (PI) are associated with higher mortality and cardiovascular diseases. Higher age is associated with a decrease of PA, only 2.4–29% of ≥60 year-olds achieve the PA times recommended by WHO. The aim of this study was to identify levels of and determinants for moderate PA, overall PA and PI in a sample of individuals aged ≥65 years.

### Methods

We analyzed baseline data from an intervention-study aiming to increase PA and decrease PI by automatically generated feedback letters to objectively measured PA and PI. Recruitment was multimodal including re-contacting participants of previous studies and advertisements in regional public buses and newspapers. At baseline, participants wore an accelerometer over a period of 7 consecutive days. PA was categorized using cut-points suggested by Freedsoon 1998 in light, moderate and vigorous physical intensity as well as physical inactivity. Potential determinants (self-efficacy, education) were measured by questionnaires or in a physical examination (BMI). Multiple linear regression models were fitted to identify determinants for PA and PI.

### Results

N = 199 persons (mean age 71.0 years (SD 4.9), 59.3% female) participated in the study. The weekly amount of overall PA for men was on average 1,821 minutes (SD 479.1), for women on average 1,929 minutes (SD 448.8). 79.7% of the women and 72.8% of the men achieved the WHO recommendation of 30 minutes moderate PA/day at baseline. The time of PI during the observation time period of 7 days was on average 4,057 minutes in men and

**Funding:** The study was funded by the Federal Ministry of Education and Research (BMBF) as a site project of the German Centre for Cardiovascular Research (DZHK) (funding sign: 81Z7400174). The funders have had no influence on the conceptualization and conduct of the study and will not have any role in the data analysis and publication of the results.

**Competing interests:** The authors have declared that no competing interests exist.

3,973 minutes in women. In males, age was found to be a significant negative determinant for overall PA (p = 0.002) and for moderate PA (p<0.001). Higher education was positively associated with higher levels of overall PA (p = 0.013) and moderate PA (p = 0.06) in men. BMI was a significant negative determinant for overall PA both in men (p = 0.039) and women (p = 0.032) as well as for moderate PA for women (p = 0.009). Only in women, not in men, self-efficacy was to be a significant positive determinant for overall PA (p = 0.020) as well as negatively associated with PI (p = 0.006).

## Discussion

The participants of our study showed high levels of PA. This is likely due to selection bias in this convenience sample. However, also levels of PI are very high and those correspond with average levels in the German population. The determinants for higher PA and lower PI differed between males and females. Thus, strategies for improving PA and decrease PI are likely different with respect to sex and should take individual factors (e.g. age, BMI) into account.

## Trial registration number

DRKS00010410 Date: 17 May 2017

## Introduction

Physical activity (PA) is defined as "any bodily movement produced by skeletal muscles that requires energy expenditure" [1]. There is strong evidence, that regular PA is a very effective, nonpharmacological and noninvasive health-promoting method [2–4]. A PA-promoting life-style is associated with a reduced risk of mortality and is correlated with improved overall health status [3]. Additionally, high levels of PA significantly decreases overall mortality by 22–34% and CVD mortality by 27–35% [5]. Recent reviews showed that regular PA is also associated with a 18–28% reduced risk of developing dementia in older adults [6, 7] and is considered as one of the proxies of the concept of cognitive reserve [8]. One of the two most significant modifiable risk factors for dementia is PI. Additionally, PA (in particular aerobic exercise) is positively associated with a less age-related reduction of organic brain matter [9].

International PA guidelines recommend for healthy individuals aged over 65 years at least 150 minutes/week of moderate PA or at least 75 minutes/week of vigorous PA, or an equivalent combination of weekly PA. In fact, there is no difference to recommendations for healthy younger individuals aged 18–64 years. PA should be performed in uninterrupted time periods (*bouts*) of a duration of at least 10 minutes [10, 11].

Data from the United Kingdom show that 19% of men and 14% of women aged 65–74 years reach the recommended level of PA [2]. A study from Norway showed that 29% of men and 25% of women aged 65–69 years meet the recommendations. In the age group 80–85 years only 7.1% of women and 3% of men reach the WHO-recommendations [12]. In the USA (National Health and Nutrition Survey, NHANES) the proportion of people aged over 60 years who achieved the recommended amount of PA was 2.4% [13]. In Germany, only 19.3% of men and 16.8% of women aged 60–69 years achieve the WHO recommendations regarding PA (self-reported via questionnaire). In the age group 70–79 years, this proportion declines to 16.5% in men and to 11.0% in women [14].

In Europe in 2015, 17,4% of the total population was 65 years or older [15]. In Germany, 21.2% of the population was 65 years or older in 2017 [16]. This proportion will further increase to 33% until 2060 [17]. Older age is associated with an higher risk for chronic diseases, multimorbidity [18–20]. Prevalences of CVD, including coronary heart disease (CHD) and stroke [21] will increase.

Low PA is one of the 10 leading risk factors for global mortality. Globally, 31.1% of the adults are insufficiently physically active [22]. A reduction of the prevalence of insufficient PA is a global target of the WHO [23, 24]. Physical inactivity (PI), defined "as any waking behaviour characterized by an energy expenditure ≤1.5 METs (metabolic equivalent of task) while in a sitting or reclining posture" (Sedentary Behaviour Research Network, 2012; Tremblay *et al.*, 2017) causes 3.2 million deaths per year worldwide, and in 2010 was estimated to be responsible for 69.3 million DALYs (disability-adjusted life years) globally [23]. Besides low levels of PA, also PI is a crucial risk factor for mortality [14, 25, 26]. Frequently interrupted PI is associated with positive effects on health status and a reduced risk for premature death [27]. Lack of PA and a high level of PI are associated with an array of non-communicable diseases (NCD) and e.g. responsible for approximately 21–25% of breast and colon cancers, 27% of diabetes cases and approximately 30% of ischaemic heart disease (prevalences)(World Health Organization, 2018). Additionally, it seems, that physical inactivity is an important preventable factor for Alzheimer's dementia [28].

Older people spent on average 8 to 9 hours a day sedentary which correspondents with 65–80% of their waking time [29]. Depending on the exact definition, the distrubution of PI across European countries ranged from 43.3% in Schweden up to 87.8% in Portugal. In Germany, 70.2% of men and 71.8% of women showed a sedentary lifestyle (low energy expenditure: <10% of the leisure-time expenditure in activities requiring ≥4 metabolic equivalents (MET) [30].

The amount of PA and PI depends on individual factors such as age, BMI, gender, education, social status and self-efficacy [31–33]. Additionally, environmental and policy factors [31, 33], weather conditions, and length of the day have an effect on the amount of PA of people [34, 35].

Overweight and obesity (BMI>30 kg/m$^2$) have a negative influence on the level of PA and PI in the elderly. In a cross-sectional study, in which 15,239 subjects were surveyed from 15 member states of the European Union it was found, that people with a low BMI (<20 kg/m$^2$) and normal BMI (20–25 kg/m$^2$) have low prevalence in PI (both genders). In contrast, people with a BMI above 30 kg/m$^2$ showed a more sedentary lifestyle [30, 36].

Higher education has a positive influence on PA and PI. Varo et al (EU study) showed, that people with primary level education were more sedentary than participants with higher levels of education (greater difference in women) [30].

Further factors that influence the amount of PA and PI are marital status, income, wellbeing, psychosocial variables such as self-efficacy, and social and cultural parameters [30, 31, 36, 37].

A report from the WHO about the prevention of non-communicable-diseases (NCD) in south-eastern European countries showed that the promotion of PA and reduction of PI are key aspects in public health efforts. Promoting physical activity through mass media was a primary goal for immediate action [38]. In addition, the Global Strategy on Diet, Physical Activity on Health [25] and the European Charter on Counteracting Obesity [39] underline the relevance of PA to fight against obesity.

To develop effective prevention strategies, adapted to the elderly, detailed information on the levels of PA and PI and their determinants are necessary. In this analysis, we assessed the levels of PA and PI and identified positive and negative determinants for PA and PI in a sample of healthy people aged ≥ 65 years.

## Materials and methods

The data for this analysis were retrieved from the baseline assessment of an intervention study with the goal to increase PA and reduce PI in elderly people with a low-threshold intervention (MOVING–*Motivation oriented intervention study for the elderly in Greifswald*) [40]. This study is a two-arm, randomized controlled trial consisting of assessment of eligibility, baseline examination, randomization, intervention (only the participants in the intervention group), and follow-up examinations at 3, 6, and 12 months after baseline. The study region was Western Pomerania, a rural area in the Northeast of Germany.

Study participants in the intervention group received two individual feedback letters containing objectively measured PA and PI times based on data from the accelerometer device captured at baseline and 3 months follow-up measurement. Feedback-letters were automatically generated in R software (version 3.3.2 or later, Lucent Technologies, Murray Hill, NJ, USA).

A comprehensive description of the study protocol for the MOVING study is published elsewhere [40].

### Inclusion of the participants

The study participants had to meet the following inclusion criteria:

- Age $\geq$ 65 years

- Ability to be physically active in daily life

  Exclusion criteria were:

- Inability to walk independently (e.g. permanent use of a wheelchair)

- Simultaneous participation in other studies addressing PA or PI

- Not accessible by telephone or cell phone

- Already fulfilling the WHO recommendations for PA (self-reported) for people $\geq$ 65 years at baseline ($\geq$ 300 minutes moderate PA per week)

  Recruitment was performed in 2017 in several ways:

- Re-contacting participants of a previously performed study [41];

- Recruitment at venues frequented by older people, e.g. senior sport hours in the public swimming pool, rehearsals of senior choirs, events at meeting places for elderly people;

- Persons contacted the study centre after reading articles about the project in regional newspapers and television, advertisements in regional buses, and flyers and posters that were distributed in GP practices, hospitals and meeting centers for elderly people.

  All participants were informed in detail about the study and had to give their written informed consent.

### Measures

All study participants received a baseline examination consisting of the assessment of blood pressure, somatometry data (body weight, waist and hip circumference) as well as sociodemographics (sex, age, education, job and partnership status, general health status). The SSA scale (self-efficacy towards physical exercise) was used to asses the participants' level of self-efficacy

The result of that scale is a sum in which higher values indicate higher self-efficacy towards PA [42].

After that, the study participants received an accelerometer device (ActiGraph GT3x-BT, Pensacola, FL, USA) which captures and records PA and PI continuously at a sampling frequency of 30 Hz over a period of seven consecutive days, starting at midnight after the baseline examination. Participants were instructed to wear the accelerometer device for seven days only during daytime on the right hip and to remove it only at bedtime and for water-based activities (e.g. showering, swimming). In addition, all study participants were asked to document their physical activities in a semi-standardized protocol for each day of the observation time. Besides the objective assessment of PA and PI, the participants were instructed to answer paper-based questionnaires to assess self-reported PA and PI with regard to the observation period of the accelerometer.

To asses self-reported PA, the International Physical Activity Questionnaire short form (IPAQ-SF) in German version was used. The IPAQ consists of seven items, addressing intensity and duration of PA in daily life over the last 7 days by self-report [43]. In addition, the German Physical Activity Questionnaire for 50+ (German PAQ 50+) was used to assess type and duration of PA in daily life by self-report [44]. PI was assessed by the Measure of Older Adults' Sedentary Time (MOST Questionnaire) in the German version [45].

Body mass index was calculated from measured body height and body weight ($1 = < 25$ kg/$m^2$, $2 \geq 25$ kg/$m^2$ and $3 = < 30$ kg/$m^2$, $\geq 30$ kg/$m^2$).

After seven days, the participants had to bring the accelerometer device and the fulfilled questionnaires back to the examination center.

## Data analysis

Data assessment and documentation were conducted based on eCRFs in an IT-supported documentation system, which is characterized by an independent operation, synchronization and monitoring. The software is based on the concept of offline clients and each staff member of the DZHK had individual login data [46].

The paper-based questionnaires were documented using the software Cardiff Teleform® (Electric Paper, Lüneburg, Germany). All questionnaires and the daily physical activity protocol contained 1-D barcodes to ensure anonymization of the study participants.

The ActiLife software (versions 6.13.2 to 6.13.3, ActiGraph Cop. ©, Pensacola, FL, USA) was used to download the PA data from the accelerometer. The raw data were calculated into 10-second epochs and saved in raw format as GT3X files. A valid measurement day was defined as a record of at least 10 hours of total wearing time. Measurements of at least 4 valid days were required to be included into data analysis. To categorize PA intensity, we used specific cut points based on Freedson et al. [47]. PA intensity was divided into sedentary (0–99 counts), light (100–1951 counts), moderate (1952–5724 counts), and vigorous (5725–9498 counts) PA. Non-wearing time was defined as having $\geq 60$ minutes of continual zero counts (range $\leq 2$ minutes between 0 and 100 counts).

Descriptive statistics were used to describe the population with respect to the levels of PA and PI. To identify determinants of PA and PI, multiple linear regression models were calculated. Dependent variables for the regression models were overall and moderate PA as well as PI. The effect of the independent variables age, BMI, education and self-efficacy was examined for all regression models. All independent variables were checked for multicollinearity using a correlation (Pearson). To include education as a determinant in the multiple linear regression models we used dummy variables. Referring to the Gauß-Markov-Theorem we analyzed the residuals and requirements of multiple linear regression like distribution and homoscedasticity.

Statistical significance was assumed for p-values <0.05. Data processing and statistical calculations were performed with IBM SPSS Statistics 25 or later (1989–2018 by IBM Corp. ©, Armonk, New York, USA.). All statistical analysis were performed based on pseudonymized data. All identifying data were separated from the project data to the earliest possible timepoint. All appointments were ensured face-to-face by the study staff.

### Ethics

The study was approved by the Ethics Committee of the University Medicine Greifswald (ethic approval protocol number BB071/16) and registered at the German Clinical Trials Register, ID: DRKS00010410.

### Results

Of 258 screened study participants, 199 could be included in the analyses (Fig 1).

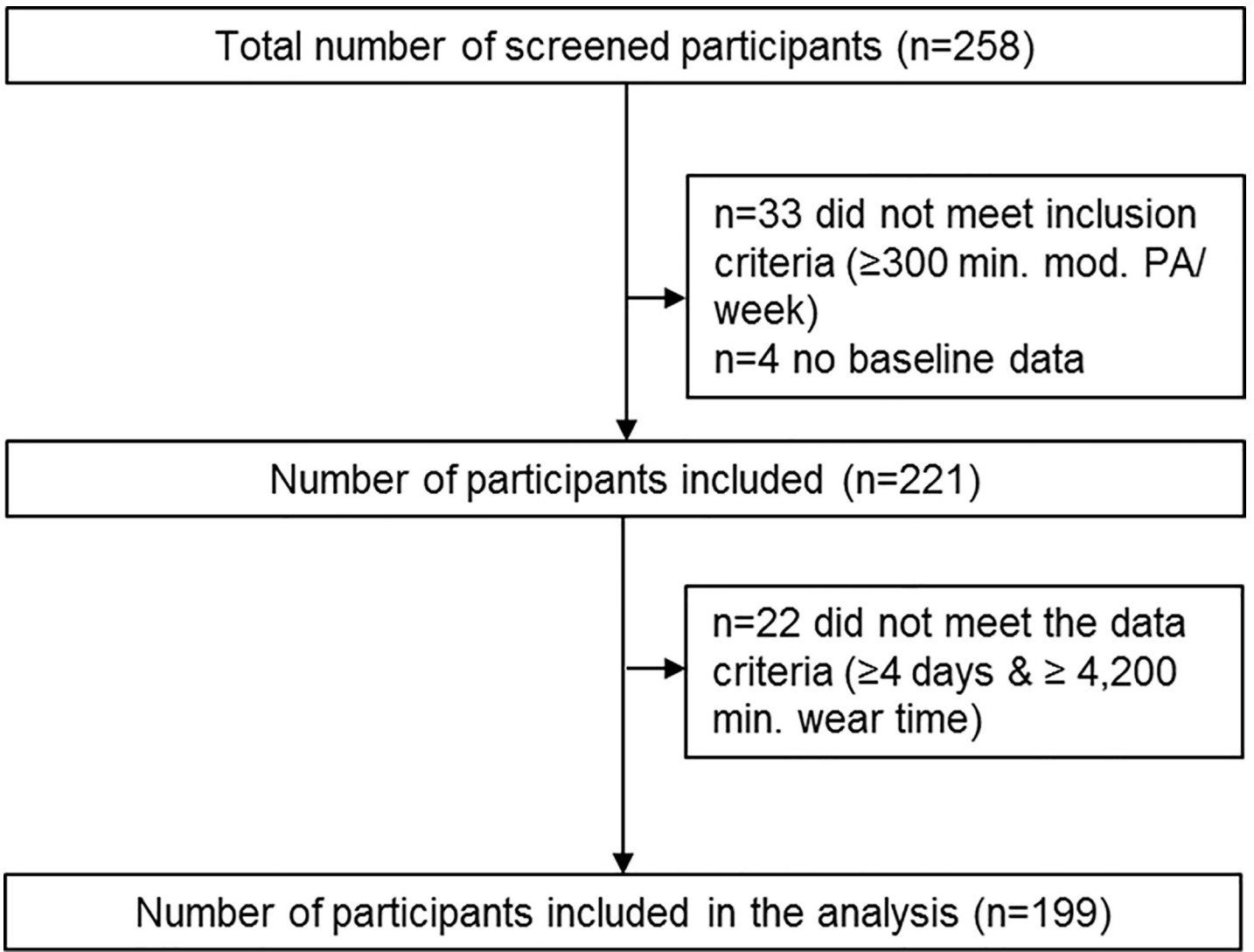

**Fig 1. Number of recruited participants and participants included in the analysis.**

**Table 1. Descriptive characteristics of the study sample (n = 199).**

| Characteristics | n | Mean (SD) or n (%) |
|---|---|---|
| Sex (women) | 199 | 118 (59.3%) |
| Age (yr) | 199 | 71.0 (SD 4.9) |
| Education (yr) | 180 | |
| < 10 years | | 38 (20.2%) |
| = 10 years | | 50 (26.6%) |
| > 10 years | | 92 (48.9%) |
| Body mass index (kg/m$^2$) | 198 | |
| < 25 | | 54 (27.3%) |
| ≥ 25 - < 30 | | 79 (39.9%) |
| ≥ 30 | | 65 (32.8%) |
| Currently Smoking (yes) | 195 | 12 (6.2%) |
| Number of participants currently having a partnership (yes) | 194 | 141 (71.9%) |
| Wearing time | 199 | 5892.8 (SD 766.6) |

*n* number of subjects, *SD* standard deviation.

199 participants were included in the analysis, thereof 118 women (59.3%) and 81 men (40.7%). The mean age was 71.0 years (SD 4.9). At the baseline measurement of PA, participants had a mean wearing time of the accelerometer of 5,892.8 minutes (SD 766.6) per week which corresponds to an average of 14.0 hours/day (Table 1).

Overall, there were no significant difference between women and men according to overall PA (mean women: 1,929.4 minutes, mean men: 1,821.2 minutes per week, T(df = 197) = 1.626; p = 0.106). Men in higher age groups showed lower levels of weekly light PA. The mean number of minutes of moderate PA increased with higher education in men and in women. Both men and women with higher BMI showed lower levels of moderate and vigorous PA (Table 2).

Male participants spent 68.9% of their waking time in PI, female participants 67.1%. Older age was associated with an increasing time of PI. In men, the level of PI increased with age (65–69 yr = 3837.3 minutes; 80+ = 4339.1 minutes) and with higher education (<10 yr = 3925.7 minutes; >10 yr = 4240.7 minutes). In women, the time of PI was largely independent of age (65–69 yr = 4,029.2 minutes; 80+ = 4,018.6 minutes) (Table 3).

N = 59 (72.8%) of the men and n = 94 (79.7%) of the women achieved the international recommendations for moderate PA (≥150 minutes moderate PA). The proportion of participants who fulfilled the recommendations decreased with age (Table 4).

A multiple linear regression was calculated to predict overall PA based on independent variables in Table 5. In men, lower age (p = 0.002), a lower BMI (p = 0.039) and higher education (p = 0.013) were found to be significant positive determinants for overall PA. Higher BMI is a negative determinant for overall PA in women (p = 0.032). Additionally, self-efficacy was found to be a significant positive determinant for overall PA (p = 0.02). The overall model fit was $R^2 = 0.23$ for men and $R^2 = 0.11$ for women (Table 5).

For moderate PA, higher age (p<0.001) was found to be a significant negative determinant in men. In women, a higher BMI was a significant negative determinant (p = 0.009). For male participants, lower education (<10 yr) was a significant negative determinant (p = 0.013) as well as 10 years of education (p = 0.006) (Table 6).

In women, better self-efficacy was found to be a significant positive determinant (p = 0.006) for PI (Table 7).

**Table 2. Mean duration of weekly PA [min.], without specific cut points, by intensity of PA and overall PA.**

| | Mean duration of weekly [min.] light PA (95% CI) | | Mean duration of weekly [min.] moderate PA (95% CI) | | Mean duration of weekly [min.] vigorous PA (95% CI) | | Mean duration of weekly [min.] overall PA (95% CI) | |
|---|---|---|---|---|---|---|---|---|
| | Male | Female | Male | Female | Male | Female | Male | Female |
| **Age (yr)** | | | | | | | | |
| 65–69 (♂ = 32; ♀ = 66) | 1602.2 (1461.0–1743.4) | 1659.7 (1566.5–1753.0) | 389.4 (324.3–454.5) | 322.8 (281.4–364.2) | 17.9 (3.5–39.2) | 5.6 (3.0–8.3) | 2009.7 (1825.0–2194.5) | 1988.9 (1884.7–2093.0) |
| 70–74 (♂ = 24; ♀ = 26) | 1453.3 (1315.5–1591.0) | 1627.7 (1502.2–1753.1) | 272.9 (197.2–348.6) | 252.8 (194.1–311.5) | 6.8 (0.3–13.9) | 4.8 (2.1–11.7) | 1733.2 (1585.6–1880.9) | 1887.9 (1736.1–2039.7) |
| 75–79 (♂ = 19; ♀ = 20) | 1511.5 (1318.3–1704.8) | 1571.9 (1319.4–1824.4) | 229.0 (150.1–307.9) | 198.0 (133.6–262.4) | 3.0 (0.4–5.6) | 1.0 (0.7–1.3) | 1743.9 (1510.6–1977.2) | 1771.0 (1491.9–2050.2) |
| 80+ (♂ = 6; ♀ = 6) | 1317.8 (924.3–1711.2) | 1654.8 (1211.0–2098.5) | 92.1 (22.8–161.4) | 322.1 (155.8–488.5) | 2.4 (0.2–4.7) | 6.3 (5.6–18.3) | 1412.4 (1004.3–1820.6) | 1983.4 (1561.5–2405.4) |
| **Education (yr)** | | | | | | | | |
| < 10 (♂ = 21; ♀ = 17) | 1489.9 (1319.5–1660.3) | 1691.7 (1445.3–1938.2) | 218.7 (166.5–271.0) | 192.0 (131.0–252.7) | 8.8 (4.6–22.2) | 1.5 (1.0–2.0) | 1717.7 (1521.1–1914.3) | 1885.3 (1622.8–2147.8) |
| = 10 (♂ = 15; ♀ = 35) | 1359.3 (1219.8–1498.9) | 1628.1 (1489.2–1767.0) | 288.8 (159.6–418.1) | 278.1 (231.6–324.6) | 27.0 (17.7–71.8) | 5.8 (2.5–9.1) | 1675.6 (1470.3–1880.9) | 1913.1 (1764.5–2061.8) |
| > 10 (♂ = 39; ♀ = 53) | 1567.8 (1440.4–1695.2) | 1632.3 (1529.2–1735.3) | 316.3 (254.0–378.6) | 317.6 (272.6–362.6) | 5.3 (1.6–9.0) | 5.8 (1.6–10.0) | 1889.6 (1727.4–2051.7) | 1957.1 (1832.4–2081.8) |
| **BMI (kg/m$^2$)** | | | | | | | | |
| <25 (♂ = 13; ♀ = 41) | 1625.6 (1313.9–1937.2) | 1663.2 (1525.1–1801.2) | 350.3 (213.2–487.5) | 331.6 (282.6–380.5) | 31.1 (21.2–83.3) | 7.4 (3.3–11.5) | 2007.0 (1646.7–2367.2) | 2003.1 (1858.0–2148.3) |
| ≥25 — <30 (♂ = 40; ♀ = 39) | 1533.8 (1417.5–1650.0) | 1698.4 (1580.0–1816.7) | 274.7 (218.7–330.7) | 315.3 (258.2–372.3) | 4.4 (0.9–7.9) | 4.4 (0.4–9.2) | 1813.2 (1664.9–1961.6) | 2019.9 (1883.3–2156.4) |
| ≥30 (♂ = 27; ♀ = 38) | 1435.3 (1308.3–1562.3) | 1547.5 (1425.2–1669.7) | 302.8 (225.5–380.3) | 207.4 (163.9–250.8) | 8.4 (1.9–18.7) | 2.1 (1.5–2.7) | 1746.7 (1571.5–1921.9) | 1757.1 (1617.3–1893.8) |
| **Currently having a partnership (yes)** | | | | | | | | |
| Yes (♂ = 70; ♀ = 71) | 1507.1 (1415.8–1598.3) | 1680.3 (1586.2–1774.4) | 283.3 (237.9–328.7) | 294.1 (254.2–334.1) | 10.1 (0.3–19.8) | 5.9 (2.6–9.3) | 1800.7 (1685.6–1915.8) | 1981.9 (1880.3–2083.6) |
| No (♂ = 8; ♀ = 45) | 1570.0 (1239.6–1900.3) | 1561.0 (1443.1–1678.9) | 356.4 (215.2–497.6) | 266.1 (220.1–313.7) | 12.2 (7.5–31.9) | 2.9 (1.2–4.6) | 1939.1 (1499.1–2379.1) | 1830.9 (1689.3–1972.5) |
| **Total** (♂ = 81; ♀ = 118) | 1515.7 (1432.4–1599.1) | 1637.5 (1565.4–1709.7) | 295.2 (253.3–337.2) | 286.2 (256.3–316.1) | 10.0 (1.4–18.5) | 4.7 (2.6–6.8) | 1821.2 (1715.3–1927.1) | 1929.4 (1847.6–2011.2) |

*M* Mean, *CI* Confidence interval, *min.* minutes.

## Discussion

The results of this analysis show that the levels of light, moderate and overall PA in our sample of older people are high. 72.8% of the male and 79.7% of the female study participants fulfilled the age specific WHO-recommendations for moderate PA for people aged over 65 years (≥150 minutes moderate PA or ≥75 minutes vigorous PA). Especially female participants in the age group +80 years were above average physically active in people in the same age [14], all women in this age group (n = 6) reached the recommendations for moderate PA.

The study results show that age is a significant negative determinant for moderate and overall PA in men. It can be concluded that men become more physically inactive with age. This is in contrast to other studies in which women are generally less physically active than their male counterparts [3, 14, 33]. This finding can also be explained by the fact that women in our sample in particular were physically active to an above-average extent.

Education was found to be a significant positive determinant regarding moderate PA in men and women and overall PA in men. These results are consistent with other studies. People

**Table 3. Mean duration of weekly PI [min.], proportion of sedentary time of total wake time in minutes.**

| | Mean min. of weekly PI (95% CI) | | Mean proportion of daily PI (daytime) in % (95% CI) | |
|---|---|---|---|---|
| | Male | Female | Male | Female |
| **Age (yr)** | | | | |
| 65–69 (♂ = 32; ♀ = 66) | 3837.3 (3633.9–4040.8) | 4029.2 (3846.7–4211.8) | 65.7% (62.8–68.5) | 66.8% (65.1–68.4) |
| 70–74 (♂ = 24; ♀ = 26) | 4166.3 (3936.2–4396.5) | 3766.2 (3514.7–4017.7) | 70.6% (68.3–72.9) | 66.4% (63.6–69.2) |
| 75–79 (♂ = 19; ♀ = 20) | 4202.4 (3736.7–4668.0) | 4042.7 (3459.1–4626.2) | 70.4% (66.4–74.3) | 68.9% (63.7–74.1) |
| 80+ (♂ = 6; ♀ = 6) | 4339.1 (3502.7–5175.6) | 4018.6 (3758.4–4278.7) | 75.3% (68.9–81.8) | 67.2% (61.9–72.4) |
| **Education (yr)** | | | | |
| < 10 (♂ = 21; ♀ = 17) | 3925.7 (3646.8–4204.6) | 3639.3 (2939.8–4338.8) | 69.4% (65.8–73.0) | 64.9% (59.4–70.4) |
| = 10 (♂ = 15; ♀ = 35) | 4047.6 (3795.2–4300.0) | 4112.8 (3908.9–4313.7) | 70.8% (67.8–73.8) | 68.3% (66.3–70.3) |
| > 10 (♂ = 39; ♀ = 53) | 4240.7 (3986.3–4495.2) | 3988.6 (3796.7–4180.5) | 69.0% (66.5–71.6s) | 67.0% (65.1–68.9) |
| **BMI (kg/m²)** | | | | |
| <25 (♂ = 13; ♀ = 41) | 3905.5 (3559.6–4251.3) | 3961.4 (3778.9–4143.9) | 66.3% (61.4–71.2) | 66.4% (64.2–68.7) |
| ≥25 - <30 (♂ = 40; ♀ = 39) | 4175.7 (3915.7–4435.7) | 3862.1 (3506.4–4217.9) | 69.5% (66.9–72.0) | 65.0% (62.5–67.5) |
| ≥30 (♂ = 27; ♀ = 38) | 3979.5 (3756.6–4202.3) | 4099.3 (3879.3–4319.4) | 69.5% (66.8–72.3) | 69.9% (67.6–72.2) |
| **Currently having a partnership (yes)** | | | | |
| Yes (♂ = 70; ♀ = 71) | 4066.2 (3893.0–4239.3) | 3873.9 (3696.8–4051.0) | 69.2% (67.3–71.1) | 65.9% (64.2–67.7) |
| No (♂ = 8; ♀ = 45) | 4045.1 (3530.3–4559.9) | 4123.9 (3853.0–4394.8) | 67.7% (61.5–74.0) | 69.0% (66.7–71.4) |
| **Total** (♂ = 81; ♀ = 118) | 4057.6 (3902.4–4212.8) | 3973.0 (3825.3–4120.7) | 68.9% (67.2–70.6) | 67.1% (65.7–68.5) |

*CI* Confidence interval, *min.* minutes.

with higher education tend to be more physically active and generally pay more attention to their personal health [30, 33].

BMI was found to be a significant determinant for moderate PA in women. Additionally, higher BMI in men and women was statistically significantly associated with a decreasing level of overall PA. Self efficacy was a significant negative determinant for PI only in women. Thus, females with higher self efficacy showed lower levels of PI. There was no systematic difference between the gender. Self efficacy was the only statistically significant predictor for PI. Thus, motivation can be seen as an important factor to reduce PI.

Our study participants spent most of their waking time in PI, men spent 68.9% and female 67.1% per day in PI, which correspondends to results from other studies examining the same age group (65% - 80%) [29, 48]. In industrialized countries, high PI values are therefore associated with an enormous burden on health systems [23].

**Table 4. Number and proportion of study participants who fulfilled the WHO recommendations for moderate PA, separately for men and women.**

| | Male | | Female | |
|---|---|---|---|---|
| | n (% of age group) | % of all men | n (% of age group) | % of all women |
| Age (yr) | | | | |
| 65–69 (n = 98) | 29 of 32 (90.6%) | 35.8% | 56 of 66 (84.8%) | 49.1% |
| 70–74 (n = 50) | 17 of 24 (70.8%) | 21.0% | 19 of 26 (73.1%) | 16.7% |
| 75–79 (n = 39) | 12 of 19 (63.2%) | 14.8% | 13 of 20 (65.0%) | 11.4% |
| 80+ (n = 12) | 1 of 6 (16.7%) | 1.2% | 6 of 6 (100%) | 5.3% |
| Total (n = 199) | 59 of 81 | 72.8% | 94 of 118 | 79.7% |

*n* number of subjects, fulfillment of WHO recommendations for moderate PA for people aged 65 and older: ≥150 minutes moderate PA or ≥75 minutes vigorous PA or an equivalent combination per week.

**Table 5. Multiple linear regression model to identify determinants for overall PA, separately for men and women.**

| Predictors | Male (n = 62, adj. $R^2$ = 0.23) | | Female (n = 91, adj. $R^2$ = 0.11) | |
|---|---|---|---|---|
| | β (95% CI) | p | β (95% CI) | p |
| Intercept | 5406.66 (3189.607–7623.706) | <0.001** | 3344.65 (1422.593–5266.706) | 0.001** |
| Age | -36.85 (-60.003––13.689) | 0.002** | -18.96 (-42.561–4.634) | 0.114 |
| BMI | -31.10 (-60.537––1.685) | 0.039* | -21.39 (-40.863––1.912) | 0.032* |
| Education[a] | | | | |
| <10 yr | -257.34 (-538.976–24.298) | 0.072 | 31.68 (-287.079–350.446) | 0.844 |
| = 10 yr | -400.08 (-712.358––87.801) | 0.013* | -56.61 (-285.765–172.546) | 0.624 |
| >10 yr ("Abitur", qualification for the university) | 62.07 (-235.838–359.973) | 0.678 | -179.96 (-470.201–110.276) | 0.221 |
| other | -52.37 (-720.459–615.722) | 0.876 | -5.99 (-546.136–534.163) | 0.982 |
| Self-efficacy | 1.35 (-7.230–9.926) | 0.754 | 9.50 (1.552–17.438) | 0.020* |

β regression coefficient, CI confidence interval

**p<0.01

*p<0.05

[a] reference: high school degree.

The participants showed a very good adherence in wearing the accelerometer device. The mean wearing time per day was 14 hours. Only 22 of 225 participants had to be excluded from the analysis because of insufficient wearing time (<4 days and <10h per day). Thus, the intervention in our study can be seen as feasible and practicable for this age group.

The public health relevance of PA and PI is high. Due to the demographic changes, maintaining high levels of PA over older ages will become even more important. Regular PA is positively associated with several physical health outcomes. Besides that, high levels of PA have also the potential to maintain cognitive health over the long term into higher ages [8, 9]. In general, high levels of PA can reduce incidence of dementia and cognitive restrictions significantly [49]. Norton et al. has shown that changeable risk factors such as PA and PI are responsible for around a third of Alzheimer's disease worldwide. In Europe, USA and UK physical inactivity was the strongest risk factor [28].

**Table 6. Multiple linear regression model to identify determinants for moderate PA, separately for men and women.**

| Predictors | Male (n = 62, adj. $R^2$ = 0.30) | | Female (n = 91, adj. $R^2$ = 0.19) | |
|---|---|---|---|---|
| | β (95% CI) | p | β (95% CI) | p |
| Intercept | 2076.30 (1235.748–2916.848) | <0.001** | 894.34 (266.772–1521.912) | 0.006** |
| Age | -19.73 (-28.505––10.946) | <0.001** | -6.09 (-13.793–1.616) | 0.120 |
| BMI | -9.71 (-20.866–1.447) | 0.087 | -8.49 (-14.849––2.131) | 0.009** |
| Education[a] | | | | |
| <10 yr | -137.18 (-243.953––30.399) | 0.013* | -99.92 (-203.994–4.164) | 0.060 |
| = 10 yr | -168.82 (-287.215––50.427) | 0.006** | -48.98 (-123.801–25.842) | 0.197 |
| >10 yr ("Abitur", qualification for the university) | -36.06 (-149.003–76.887) | 0.525 | -61.84 (-156.602–32.929) | 0.198 |
| Other | 54.59 (-198.699–307.887) | 0.667 | 175.51 (-0.851–351.877) | 0.051 |
| Self-efficacy | -0.34 (-3.590–2.914) | 0.836 | 1.49 (-1.102–4.084) | 0.256 |

β regression coefficient, CI confidence interval

**p<0.01

*p<0.05

[a] reference: high school degree.

**Table 7. Multiple linear regression model to identify determinants for PI, separately for men and women.**

| Predictors | Male (n = 62, adj. $R^2$ = 0.03) | | Female (n = 91, adj. $R^2$ = 0.14) | |
| --- | --- | --- | --- | --- |
| | β (95% CI) | p | β (95% CI) | p |
| Intercept | 1222.26 (-2540.348–4984.866) | 0.518 | 5684.80 (2881.732–8487.870) | <0.001** |
| Age | 36.92 (-2.380–76.220) | 0.065 | -16.49 (-50.903–17.924) | 0.343 |
| BMI | 6.18 (-43.762–56.117) | 0.805 | 13.05 (-15.355–41.451) | 0.364 |
| Education[a] | | | | |
| <10 yr | -217.33 (-695.297–260.647) | 0.366 | -477.20 (-942.076–−12.329) | 0.044 |
| = 10 yr | 6.97 (-523.010–536.940) | 0.979 | 38.34 (-295.850–372.538) | 0.820 |
| >10 yr ("Abitur", qualification for the university) | 20.14 (-485.440–525.725) | 0.937 | 196.84 (-226.431–620.120) | 0.358 |
| Other | -1351.47 (-2485.301–−217.637) | 0.020** | -667.82 (-1455.558–119.918) | 0.096 |
| Self-efficacy | 3.09 (-11.468–17.648) | 0.672 | -16.30 (-27.881–−4.713) | 0.006** |

β regression coefficient, CI confidence interval

**p<0.01

*p<0.05

[a] reference: high school degree.

Effective and practicable strategies to increase PA and decrease PI are needed especially for the elderly. A report from WHO for the European Region mentions specific suggestions to increase peoples physical activity (e.g. by promotion of green spaces or cycle paths etc.) [39]. Therefore, to design, scale, and implement effective non-communicable disease prevention programmes, accurate and valid data on physical activity levels and on sedentary behavior are needed as well as valid knowledge about significant determinants of both PA and PI.

Regular PA is a highly effective health-promoting method with strong evidence [2, 4, 23, 25]. International recommendations for PA are existing for all age groups [10, 50], but it still remains under debate how people should accumulate their recommended time of PA over the week [51].

In recent years, the number of interventions targeting PI has increased. However, there is still a lack of recommendations for PI [52]. In addition, it is also not conclusively established whether PI is an independent risk factor for chronic diseases [52]. In summary, recent literature points out that public health activities should emphasise increasing PA at any intensity especially in the elderly [53].

## Limitations

This analysis is based on a convenience sample of participants. We used a variety of sampling methods, including the possibility of self-recruitment, some of which have likely increased the proportion of participants with above average PA compared to PA at the population level. We observed high levels of PA, particularly among the older age groups and females which indicate some selection bias.

In general, the use of accelerometer devices for objective assessment of PA allows a valid and reliable record of PA intensity, frequency, and duration. But data from the accelerometer device can potentially differ from the real levels of PA and PI especially in the elderly because several activities are carried out in standing (e.g. gardening) or sedentary positions (e.g. gymnastic on stools) which, as a consequence, can not be assessed reliably.

In general, the explanatory value for both PA models are acceptable. The explanatory value for our PI model is low. Thus, research should focus on further environmental and interpersonal factors, e.g the walkability of neigbourhoods and attractive activities for seniors.

## Conclusion

The number of guidelines and recommendations in PA and PI increases continuously. However, there are still certain aspects especially of PI which can not described in an accurate way. To our knowledge we still know only little about the independent negative health effects of sedentary behavior. In general, there is international consensus regarding recommendations for PA but recommendations about PI are still under debate.

Because of demographic change and the associated increase of proportion of older people the need of global prevention strategies is high, simultaneously knowledge to improve our understanding of PA and PI in the elderly. Prevalence of dementia and chronic diseases (e.g. CVD) will probably increase in the next decades. Therefore, the relevance of modifiable risk factors as preventive measures will rise [49].

Our analyses confirm, that especially individual factors (e.g. age, sex) has the largest impact on PA. The results for PI are less clear. Thus, PI may have other predictors. Another important finding of this study is that PA and PI can be seen as mostly independent factors of activity. Participants with a high level of PA showed also high levels of PI. Further research should pay more attention to effective predictors for the reduction of PI and should focus more on environmental and interpersonal factors especially in PI.

## Supporting information

**S1 File. Dataset.**
(XLSX)

## Author Contributions

**Conceptualization:** Fabian Kleinke, Sabina Ulbricht, Marcus Dörr, Wolfgang Hoffmann, Neeltje van den Berg.

**Data curation:** Neeltje van den Berg.

**Formal analysis:** Fabian Kleinke.

**Methodology:** Wolfgang Hoffmann, Neeltje van den Berg.

**Project administration:** Fabian Kleinke.

**Software:** Peter Penndorf.

**Supervision:** Neeltje van den Berg.

**Writing – original draft:** Fabian Kleinke.

**Writing – review & editing:** Sabina Ulbricht, Marcus Dörr, Wolfgang Hoffmann, Neeltje van den Berg.

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
