## [Decision Letter · Decision Letter 0]

7 Jan 2020

PONE-D-19-30791

Levels of and determinants for physical activity and physical inactivity in a group of healthy elderly people in Germany: Baseline results of the MOVING-study

PLOS ONE

Dear Mr Kleinke,

Thank you for submitting your manuscript to PLOS ONE. After careful consideration, we feel that it has merit but does not fully meet PLOS ONE’s publication criteria as it currently stands. Therefore, we invite you to submit a revised version of the manuscript that addresses the points raised during the review process.

The reviewer has major concerns about the statistical analyses and therefore these should be addressed.

We would appreciate receiving your revised manuscript by Feb 21 2020 11:59PM. To enhance the reproducibility of your results, we recommend that if applicable you deposit your laboratory protocols in protocols.io, where a protocol can be assigned its own identifier (DOI) such that it can be cited independently in the future. For instructions see: http://journals.plos.org/plosone/s/submission-guidelines#loc-laboratory-protocols

We look forward to receiving your revised manuscript.

Kind regards,

Martin Senechal, PhD

Academic Editor

PLOS ONE

Journal Requirements:

2. Please correct your reference to "p=0.000" to "p<0.001" or as similarly appropriate, as p values cannot equal zero.

3. At line 151, please provide a reference for the "previously performed study" mentioned.

4. Please upload a copies of Supporting Information Tables S1-S7  and Supporting Information  Figure S1 which you refer to in your lines 500- 501 .

Reviewers' comments:

Reviewer's Responses to Questions

**Comments to the Author**

1. Is the manuscript technically sound, and do the data support the conclusions?

Reviewer #1: Partly

2. Has the statistical analysis been performed appropriately and rigorously? 

Reviewer #1: No

3. Have the authors made all data underlying the findings in their manuscript fully available?

Reviewer #1: Yes

4. Is the manuscript presented in an intelligible fashion and written in standard English?

Reviewer #1: Yes

5. Review Comments to the Author

Reviewer #1: The manuscript addresses an interesting topic. The data are unique and the statistical modelling employed, though rather standard, is sound. The results are promising, but there are a number of major concerncs which must be addressed to ensure the reliability of the results.

1. I really appreaciate that the data are attached to the manuscript. This allows me to replicate some of the analysis. However, the number of observations in the .xls file is different from the one reported in the main text. This is a major issue as it is rather unclear which is tha database used for the final analysis. Please, amend the data accordingly.

2. The use of multiple regression is rather sound, though more sophisticated methods could be used. My major concer is about the obtained inferential results. The regression model, as well as any other parametric test, must fulfill some crucial assumptions. For the regression model, you should refer to the Gauss-Markov theorem. If any of the model's assumptions are not fulfilled, then the results are non reliable and modifications of the basic model must be provided. A clear and deep analysis of the residuals must be included. Heteroskedasticity, outlying observations, skewness and heavy-tails may strongly affect your results, making model inference completely misleading. From a simple graphical inspection of the residuals, all these aspects can be easily detected.

3. The goodness-of-fit of the defined regressions is rather poor. This limits the use of those models for the prediction of the response variable, given the covariates. Please, investigate further model specifications, may be including interaction terms and/or latent variables. A more detailed descriptive and graphical analysis of the data may reveal interesting relationshios between the response and the covariates.

6. PLOS authors have the option to publish the peer review history of their article (what does this mean?). If published, this will include your full peer review and any attached files.

Reviewer #1: No

---

## [Author Response · Author response to Decision Letter 0]

20 Feb 2020

Please see graphics attached in the uploaded file "Response to Reviewers"

File Inventory Manuscript Number: PONE-D-19-30791

Reviewer note #1

The manuscript addresses an interesting topic. The data are unique and the statistical modelling employed, though rather standard, is sound. The results are promising, but there are a number of major concerncs which must be addressed to ensure the reliability of the results.

I really appreaciate that the data are attached to the manuscript. This allows me to replicate some of the analysis. However, the number of observations in the .xls file is different from the one reported in the main text. This is a major issue as it is rather unclear which is tha database used for the final analysis. Please, amend the data accordingly.

Anwser #1

Thank you very much for your constructive comments to the manuscript. In our opinion, your comments will help to increase the quality of the manuscript, especially the part of the statistical analyses. 

First, the attached data file included all study participants (n=221) without excluding participants who do not meet the inclusion criteria (n=22). We removed these participants from the attached file. The updated file contains data of n=199 participants. We calculated our results with n=199 participants. 

Reviewer note #2

2. The use of multiple regression is rather sound, though more sophisticated methods could be used. My major concer is about the obtained inferential results. The regression model, as well as any other parametric test, must fulfill some crucial assumptions. 

For the regression model, you should refer to the Gauss-Markov theorem. If any of the model's assumptions are not fulfilled, then the results are non reliable and modifications of the basic model must be provided. A clear and deep analysis of the residuals must be included. Heteroscedasticity, outlying observations, skewness and heavy-tails may strongly affect your results, making model inference completely misleading. From a simple graphical inspection of the residuals, all these aspects can be easily detected.

Answer #2

Thank you for your constructive comment to the analyses of the residuals. 

For the regression model, we added the reference to the Gauss-Markov theorem. We analyzed the residuals. You can find the residuals in the attached dataset “S1_dataset.xlsx” (variables: RES_1_overall_PA, PRE_1_overall_PA, RES_1_moderate_PA, PRE_1_moderate_PA, RES_1_PI, PRE_1_PI). 

We included a detailed analysis of the residuals. First, we graphically analyzed the residuals (see attachments). We came to the result, that all tested residuals are normally distributed. The attached graphics and g-g-plots show that all tested residuals are between range -3 and 3 SD. In addition, no systematic patterns or only very light deviations were detected. Please see also graphical inspection of the residuals: 

GRAPHIS (see attachments)

In addition to the graphical verification, we analyzed the distribution of the residuals also with the Shapiro-Wilk-Test. In these analyses, we came to the same results. All tested residuals are (nearly) normally distributed. Furthermore, literature points out that a small deviation of the normal distribution in case of linear regression is acceptable, especially in bigger samples (Lumley et al. 2002), Eid, 2010). Even if the residuals are not unequivocally normally distributed, this does not lead to a distortion of the regression coefficient (Eid, 2010). In summary, the analyses showed that the assumption of homoscedasticity is given. No systematic deviations, strong skewness or heavy tails were detected.

In principle, an uneven distribution of the residuals can lead to a distortion of the standard error, which possibly can distort the significance test. To verify the results of the significance test, we calculated a bootstrapped regression model (Field, 2008). We performed a bootstrapped regression with 1000 repetitions (Chernick, 2008). In these analyses, we came to the same results: the same predictors became significant. In sum, the results of our multiple linear regression models are reliable.

In addition, we performed Durbin-Watson test to check uncorrelation of the residuals. The results of the Durbin-Watson test should be close to 2. By checking the uncorrelation it was found that this requirement is fulfilled. Additionally, we tested multicollinearity between the variables (results should be smaller than 5) and analyzed the condition index. Results of the condition index should be smaller than 30. In summary, all requirements are fulfilled and results of the linear regression models are reliable. 

We added the following text to the manuscript (page 9): “Referring to the Gauß-Markov-Theorem we analyzed the residuals and requirements of multiple linear regression like distribution and homoscedasticity”.

Reviewer note #3

The goodness-of-fit of the defined regressions is rather poor. This limits the use of those models for the prediction of the response variable, given the covariates. Please, investigate further model specifications, may be including interaction terms and/or latent variables. A more detailed descriptive and graphical analysis of the data may reveal interesting relationshios between the response and the covariates.

Answer #3

Thank you very much for this comment. 

The chosen predictors explain up to 30% of the variance of the model (table 6). Only the model of physical inactivity (PI) shows an unsatisfactory explanatory value for male participants (R2=0.03). Obviously, this is because other predictors, especially in men, affect physical inactivity. This result is in line with current research, that physical activity and physical inactivity can be seen as independent factors (Stamatakis, 2018). Furthermore, this study examines human behavior that is often influenced by other, not measurable factors. In analyzes of human behavior, lower R2 values (>50%) can be expected. 

We defined possible determinants for PA and PI based on literature analysis and hypothesis. A priori, no interaction terms were defined. In our opinion, a retrospective examination of interaction terms is not purposeful. In case of retrospective examination, there is a possibility that results are random and not based on hypothesizes. In conclusion, only a limited interpretation of these results would be possible. However, we are grateful for this note and we will consider this aspect for future analyses. 

 

References

Eid, M., Gollwitzer, M. & Schmitt, M. (2010). Statistik und Forschungsmethoden. Lehrbuch. Mit Online-Materialien. Weinheim: Beltz.

Cohen, J. (1988). Statisitcal power analyses for behavioral sciences (2nd ed). Hillsdale, NJ: Erlbaum Associates.

Field, A. (2009). Discovering statistics using SPSS (3rd ed). London, SAGE 

Michael R. Chernick (2008) Bootstrap methods: a guide for practioners and researchers, 2. Auflage, Hoboken, NJ: Wiley.

Lumley, Thomas, Paula Diehr, Scott Emerson, and Lu Chen. 2002. “The Importance of the Normality Assumption in Large Public Health Data Sets.” Annual Review of Public Health 23(1):151–69.

---

## [Decision Letter · Decision Letter 1]

8 Jun 2020

PONE-D-19-30791R1

Levels of and determinants for physical activity and physical inactivity in a group of healthy elderly people in Germany: Baseline results of the MOVING-study

PLOS ONE

Dear Dr. Kleinke,

Thank you for submitting your manuscript to PLOS ONE. After careful consideration, we feel that it has merit but does not fully meet PLOS ONE’s publication criteria as it currently stands. Therefore, we invite you to submit a revised version of the manuscript that addresses the points raised during the review process.

We look forward to receiving your revised manuscript.

Kind regards,

Thalia Fernandez, Ph.D.

Academic Editor

PLOS ONE

Additional Editor Comments (if provided):

Dear Dr Kleinke,

Although the reviewer who viewed the original manuscript says that the article must be accepted, I sent the revised manuscript to another reviewer, and she found some details. I agree with her that the discussion section should include more interpretive work.

I also think that phrases such as "Overall, women were slightly more physically active than male participants (mean 1,929.4 min 228 and 1,821.2 min per week, respectively)" include some degree of imprecision because the authors did not use any statistical analysis to make these comparisons; therefore, they found no significant differences (in this case, between genders). I recommend using statistical analyzes of comparison of means (and consider the multiplicity of tests to adjust its level of significance).

Perhaps the Introduction, and especially the Discussion, could be enriched if the physical activity were considered as one of the proxies of the Cognitive Reserve (Stern, 2009). Aging is not only related to cardiovascular disease; aging is the principal risk-factor to MCI and dementia. Therefore, I believe that the authors will agree that, in aging, it is not only desirable that the cardiovascular system remains healthy, but also cognitive functions.

Also, author should check the abbreviation for "minutes".

Reviewers' comments:

Reviewer's Responses to Questions

**Comments to the Author**

1. If the authors have adequately addressed your comments raised in a previous round of review and you feel that this manuscript is now acceptable for publication, you may indicate that here to bypass the “Comments to the Author” section, enter your conflict of interest statement in the “Confidential to Editor” section, and submit your "Accept" recommendation.

Reviewer #1: All comments have been addressed

Reviewer #2: All comments have been addressed

2. Is the manuscript technically sound, and do the data support the conclusions?

Reviewer #1: (No Response)

Reviewer #2: Partly

3. Has the statistical analysis been performed appropriately and rigorously? 

Reviewer #1: (No Response)

Reviewer #2: I Don't Know

4. Have the authors made all data underlying the findings in their manuscript fully available?

Reviewer #1: (No Response)

Reviewer #2: Yes

5. Is the manuscript presented in an intelligible fashion and written in standard English?

Reviewer #1: (No Response)

Reviewer #2: Yes

6. Review Comments to the Author

Reviewer #1: (No Response)

Reviewer #2: The manuscript has problems in organizing the topics in introduction section. Information is missing in the material and methods section and Discussion section needs more interpretative work.

7. PLOS authors have the option to publish the peer review history of their article (what does this mean?). If published, this will include your full peer review and any attached files.

Reviewer #1: No

Reviewer #2: No

---

## [Author Response · Author response to Decision Letter 1]

22 Jun 2020

Abstract 

1. This line is confusing, rewrite using the terms light, moderate and vigorous 

“Insufficient physical activity (PA) and high levels of physical inactivity (PI)” 

Thank you for this comment. In order to avoid confusions and to structure the statement more clearly, we have adapted the sentence. “Low levels of PA and high levels of physical inactivity (PI) are associated with higher mortality and cardiovascular diseases.” 

Introduction

1. The introduction does not have a clear structure. I suggest the following structure:

Manuscript

• Effect of insufficient PA

• Definition PI or sedentary behavior

• Effect of insufficient PA and PI

• Effect of regular PA

• General prevalencia (disease)

• Recomendations (moderate PA)

• Global prevalence (recommendations moderate PA)

• Germany prevalence (recommendations moderate PA)

• Prevalence sedentary 

• Factors affecting amount of PA and PI

• Prevention

 Suggestion

• Definition of physical activity (PA)

• Effect of regular PA

• Recommendations (moderate PA)

• General prevalence (recommendations moderate PA)

• Germany prevalence (recommendations moderate PA)

• General prevalencia (disease)

• Definition PI or sedentary behavior

• Effect of insufficient PA and PI

• Prevalence sedentary 

• Factors affecting amount of PA and PI

• Prevention

Thank you for your comment to the structure of the introduction. We totally accepted your comment and structured the introduction section accordingly your suggestion. We marked all changes with yellow color. Now, the introduction part is structured more clearly. We rebuilt the introduction section, restructured, and separated some parts and sentences. Therefore, it might be laborious to track all the changes we have made. 

2. The use of definitions is not clear, please use light PA, moderate PA and vigorous PA or low or high levels of PA in the introduction section.

We tried to make the introduction section easier to understand by using the terminology low and high levels of PA and PI and by following your recommendations to structure. We changed all relevant parts in the introduction section (for example lines 64, 90 or 97). 

3. Explained in detail the background studies in introduction section. 

“Overweight and obesity (BMI>30 Kg/m2) have an negative influence on the level of PA and PI in the elderly. Higher education has a positive influence on PA. Further factors that influence the amount of PA and PI are marital status, income, wellbeing, psychosocial variables such as self-efficacy, and social and cultural parameters (29,30,35,36)

In this section, we report which determinants have an influence on the amount of PA and PI. We added further background information to the referenced studies. For example, please see lines 115-119: “Overweight and obesity (BMI>30 kg/m2) have a negative influence on the level of PA and PI in the elderly. In a cross-sectional study, in which 15,239 subjects were surveyed from 15 member states of the European Union, it was found, that people with a low BMI (<20 kg/m2) and normal BMI (20-25 kg/m2) have low prevalence in PI (both genders). In contrast, people with a BMI above 30 kg/m2 showed a more sedentary lifestyle (30,36).”

A report from the WHO about the prevention of non-communicable-diseases (NCD) in south eastern European countries showed that the promotion of PA and decrease reduction of PI are key aspects in public health efforts (37)”

We expanded information about this topic and added the following sentences to the introduction section (see lines 128-131): “Promoting physical activity through mass media was a primary goal for immediate action (38). In addition, the Global Strategy on Diet, Physical Activity on Health (25) and the European Charter on Counteracting Obesity (39) underline the relevance of PA to fight against obesity.”

Materials and Methods

1. Rewrite the following paragraph including the variables considered in the regression, independent and dependent variables. 

“Descriptive statistics were used to describe the population with respect to the levels of PA and PI. To identify determinants of PA and PI, multiple linear regression models were calculated. All variables were checked for multicollinearity using a correlation (Pearson)”

Thank you for this comment to the method section. We added the following sentences to the manuscript (lines 216-218): “Dependent variables for the regression models were overall and moderate PA as well as PI. The effect of the independent variables age, BMI, education and self-efficacy was examined for all regression models. All independent variables were checked for multicollinearity using a correlation (Pearson). ”

Discussion

1-Organize the discussion section based on levels of PA. 

Discussion section requires more work; in particular, you must interpret the models you obtained from each regression, then you will be able to compare your interpretations with previous studies.

Thank you for your comment to the structure of the discussion section. To address your suggestion, we restructured some parts in the discussion based on intensity of PA levels (moderate PA, overall PA and PI). Therefore, we rebuilt the discussion section, restructured, and separated some parts and sentences (please see yellow marks). 

In addition, we followed your recommendation and expanded the discussion section on interpretation and information from other studies. For example (lines 290-294): “The study results show that age is a significant negative determinant for moderate and overall PA in men. It can be concluded that men become more physically inactive with age. This is in contrast to other studies in which women are generally less physically active than their male counterparts (3,14,33). This finding can also be explained by the fact that women in our sample in particular were physically active to an above-average extent.”

2. Why is this relevant to the discussion?, this is information should be included in material and methods section.

“The participants showed a very good adherence in wearing the accelerometer device. The mean wearing time per day was 14 hours. Only 22 of 225 participants had to be excluded from the analysis because of insufficient wearing time”

Thank you for your comment. In our opinion, the finding about the wearing time is important and relevant. In addition to the effect, the feasibility and practicability of the intervention are important for the success of the study. The good wearing time of the accelerometer device shows, that such interventions are suitable for older people. This means that further studies can be carried out using the same method without major concerns or adjustments in methods. Therefore, we added the following sentence to the manuscript (see lines 311-312): “Thus, the intervention in our study can be seen as feasible and practicable for this age group.”

3. Did you make any intervention in this study? Your participants got feedback about their PA, but you do not have a control measure before wear the accelerometer. I think this paragraph should be eliminated.

“In recent years, the number of interventions targeting PI has increased. However, there is still a lack of recommendations for PI (49). In addition, it is also not conclusively established whether PI is an independent risk factor for chronic diseases (49). In summary, recent literature points out that public health activities should emphasise increasing PA at any intensity especially in the elderly. (50)”

This section primary refers to the theoretical background and is not related to results of our intervention or study. Although number of interventions in PI are increasing in general, there are still no global recommendations for PI. Therefore, we would like to keep this section in the manuscript as an overall view about the topic and relevance. 

Minor suggestions:

Thank you for carefully reviewing our manuscript. We implemented all minor comments. 

Doble parentheses líne 63

Misplaced reference line 66 (7,8)

Error in parentheses líne 69

Doble parentheses line109

Error in word “therof” líne 221

---

## [Decision Letter · Decision Letter 2]

29 Jul 2020

Levels of and determinants for physical activity and physical inactivity in a group of healthy elderly people in Germany: Baseline results of the MOVING-study

PONE-D-19-30791R2

Dear Dr. Kleinke,

We’re pleased to inform you that your manuscript has been judged scientifically suitable for publication and will be formally accepted for publication once it meets all outstanding technical requirements.

Kind regards,

Thalia Fernandez, Ph.D.

Academic Editor

PLOS ONE

Reviewers' comments:

Reviewer's Responses to Questions

**Comments to the Author**

1. If the authors have adequately addressed your comments raised in a previous round of review and you feel that this manuscript is now acceptable for publication, you may indicate that here to bypass the “Comments to the Author” section, enter your conflict of interest statement in the “Confidential to Editor” section, and submit your "Accept" recommendation.

Reviewer #1: All comments have been addressed

Reviewer #2: All comments have been addressed

2. Is the manuscript technically sound, and do the data support the conclusions?

Reviewer #1: (No Response)

Reviewer #2: Yes

3. Has the statistical analysis been performed appropriately and rigorously? 

Reviewer #1: (No Response)

Reviewer #2: Yes

4. Have the authors made all data underlying the findings in their manuscript fully available?

Reviewer #1: (No Response)

Reviewer #2: Yes

5. Is the manuscript presented in an intelligible fashion and written in standard English?

Reviewer #1: (No Response)

Reviewer #2: Yes

6. Review Comments to the Author

Reviewer #1: (No Response)

Reviewer #2: The authors have made a great effort to follow the recomendations improving the structure and content of the paper, however the discussion section needs more work. They have described the factos that are associates with increased physical activity, I would expect in discussion section an explanation about why these factors and no others affect the physical activity.

7. PLOS authors have the option to publish the peer review history of their article (what does this mean?). If published, this will include your full peer review and any attached files.

Reviewer #1: No

Reviewer #2: No

---

## [Editor Report · Acceptance letter]

4 Aug 2020

PONE-D-19-30791R2 

Levels of and determinants for physical activity and physical inactivity in a group of healthy elderly people in Germany: Baseline results of the MOVING-study 

Dear Dr. Kleinke:

I'm pleased to inform you that your manuscript has been deemed suitable for publication in PLOS ONE. Congratulations! Your manuscript is now with our production department. 

Kind regards, 

on behalf of

Dr. Thalia Fernandez 

Academic Editor

PLOS ONE